# Cellular transcriptomics of arrested normal lung fibroblasts IMR-90 infected with Human Adenovirus 5 E1A mutants

Rafe Helwer[1], Peter Pelka[1,2]*

1 Department of Microbiology, University of Manitoba, 2 Department of Medical Microbiology and Infectious Diseases, University of Manitoba

* peter.pelka@umanitoba.ca

## Abstract

Induction of S-phase is paramount to the replication of most human DNA viruses. Human adenoviruses have evolved sophisticated mechanisms that drive the infected cells into S-phase to ensure that viral genomes are efficiently replicated. We have identified an E1A mutant, E1A289R*dl*2–11/YC, that disrupts the canonical means of S-phase induction by E1A. Specifically, this mutant abrogates binding of E1A to the E2F/DP complex as well as to the retinoblastoma protein. Yet, we show that this mutant can still effectively drive the infected cell into S-phase. We explore potential mechanisms of how this occurs via cellular transcriptomic analysis 16 hours after infection. We show that this mutant induces many cell-cycle specific genes to drive S-phase. Interestingly, MYC mRNA is significantly upregulated by this mutant as compared to other viruses investigated. This MYC upregulation, together with normal expression of E4orf6/7 in this mutant, may contribute to efficient S-phase induction. We also demonstrate that this mutant is unable to effectively suppress innate immune response to infection, likely due to loss of p300/CBP binding caused by deletion of E1A residues 2 to 11.

## Introduction

Adenoviruses are a species of non-enveloped eukaryotic viruses with double-stranded DNA genomes [1]. These viruses infect a variety of tissues and cell types and in humans generally cause a mild infection except for those who are immunocompromised [1]. Since these viruses infect terminally differentiated cells of the epithelium in most cases, their replication would be severely stifled if they were not able to drive these cells into S-phase in order to generate enough substrates and co-factors for dNTP synthesis and replication of the viral genome [2]. Consequently, adenoviruses have evolved several different mechanisms that drive cells into S-phase, activating cellular replication machinery, which the viruses then hijack to replicate their own genomes [2].

**Data availability statement:** RNA-seq data has been deposited in the GEO repository under accession # GSE288390. The records are accessible at: https://www.ncbi.nlm.nih.gov/geo/query/acc.cgi?acc=GSE288390.

**Funding:** This work was supported by grants from the Natural Sciences and Engineering Research Council of Canada to PP (Grant number: RGPIN/05366-19) and the Canadian Institutes of Health Research to PP (Grant number: PJT-173376). Funding for open access charge: CIHR PJT-173376. RH was supported with a studentship from Research Manitoba. The funders had no role in study design, data collection and analysis, decision to publish, or preparation of the manuscript.

**Competing interests:** The authors have declared that no competing interests exist.

Control of S-phase entry and cell cycle re-entry is tightly governed by multiple checkpoint mechanisms to ensure that aberrant or unscheduled genome replication does not happen (reviewed by [3]). Adenoviruses have evolved several different mechanisms that ensure that required checkpoints are bypassed so that the infected cells enter S-phase [2]. Amongst these, deregulation of the retinoblastoma (Rb) family of proteins by Early region protein 1A (E1A) was the first example showing an oncogene interfering with the function of a tumour suppressor protein (Rb) [4]. E1A disrupts the function of Rb via two regions located in conserved region (CR) 1 and CR2 of E1A (Fig 1A), which results in the release of Rb-associated E2F and transcriptional activation of cell cycle specific genes [5–9]. There are 5 isoforms of E1A and at early times after infection the most abundant isoforms are 289 amino acid residues (R) and 243R [10]. E1A243R differs from E1A289R solely in that it is lacking the CR3 region (Fig 1A) and hence 243R is deficient in transactivation of viral promoters [10]. In addition to E1A, other viral proteins contribute to the deregulation of

A

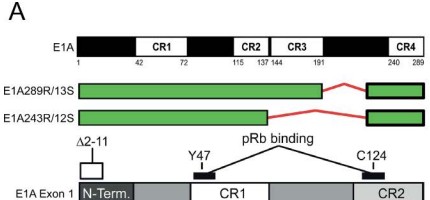

B

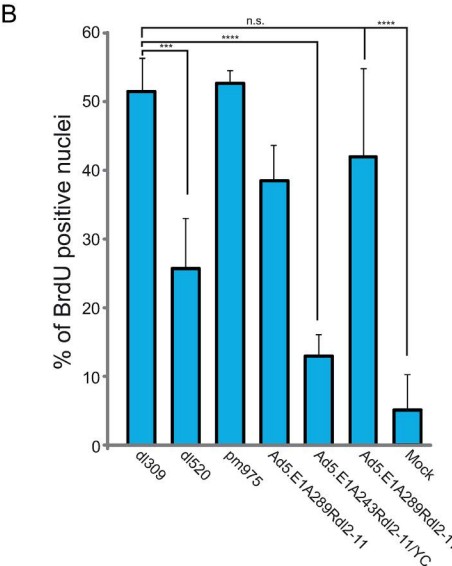

**Fig 1. E1A289R*dl*2-11/YC mutant induces S phase.** A. Schematic representation of E1A isoforms and E1Adl2-11/YC mutant. Numbers are nucleotides. The E1A289Rdl2-11/YC mutant deletes residues 2-11 and mutates Y47 and C124 and as a result is deficient at binding pRb and E2F, and viral growth. B. BrdU Assay. Arrested IMR-90 cells were infected with the indicated adenoviruses, pulsed with BrdU for 1 h at 15 hours after infection, fixed, and stained with anti-BrdU anti-body. Results are expressed as a percentage of BrdU-positive nuclei and constitute a mean of three random fields of view at a low magnification. Statistical analysis was performed using pairwise Student's t-test.

the cell cycle in infected cells. For example, E4orf6/7 is able to functionally compensate for loss of E1A and drive release of E2Fs from Rb during infection [11,12]. Additionally, E4orf6/7 is capable of inducing expression from E2F-regulated promoters by recruiting activating E2F1 to these promoters in an E1A-independent fashion [13,14]. E4orf6/7 has also been shown to drive E2F4 into the nucleus during infection [15]. Furthermore, E4orf6 forms a ubiquitin ligase complex together with E1B55k, which is able to mimic the functions of E1A in driving S-phase when E1A is absent [16,17]. Lastly, we have previously shown that E1A itself directly binds to the repressive E2Fs, particularly E2F4 and drives gene activation from repressed E2F promoters by competing for Rb binding, converting a repressive E2F4 into an activating one [2]. Collectively, these results highlight the importance of driving cells into the S-phase and the various mechanisms that adenoviruses employ to initiate and ensure their replication.

In the present study we investigated the cell cycle induction capabilities of human adenovirus (HAdV) mutants expressing E1A289R that has lost its ability to interact with the Rb family of proteins as well as E2Fs (referred to as E1A289R*dl*2–11/YC). As expected, we observed that E1A243R*dl*2–11/YC, unable to bind to Rb or E2Fs, was largely deficient for S-phase induction in arrested primary lung fibroblasts. Unexpectedly, E1A289R*dl*2–11/YC, which carried the same mutations, was still able to drive the cells into S-phase. To better understand the molecular mechanisms underlying this phenotype, we undertook transcriptomic analysis of human primary lung fibroblasts (IMR-90) infected with viruses expressing wild-type (wt) E1A (*dl*309), E1A243R (*dl*520), E1A289R (*pm*975), or E1A289Rdl2–11/YC (Ad5.E1A289R*dl*2–11/YC).

## Materials and methods

### Viruses

Ad5.E1A289R*dl*2–11/YC was generated in-house by homologous recombination as previously described [2]. Additionally, HAdV *dl*309, *dl*520, and *pm*975 were previously described [18–20].

### Cell culture and virus infections

IMR-90 (ATCC# CCL-186) cells were grown in Dulbecco's Modified Eagle's Medium (MilliporeSigma) supplemented with 10% fetal bovine serum, streptomycin and penicillin (Corning). All virus infections were carried out at a multiplicity of infection (MOI) of 10 in serum-free medium for 1 hour after which fresh complete media was added without removal of the infection media. Mock-infected cells were treated the same as those infected but with no virus added to the serum-free medium.

### BrdU incorporation assay

BrdU incorporation assay was performed as previously described without any changes [2].

### Western blot

Contact-inhibited IMR-90 cells were infected with HAdV E1A mutants at an MOI of 10 for 16 hours, after which the cells were collected by scraping them off the plates and lysed in NP-40 lysis buffer (10 mM Tris pH 7.8, 150 mM NaCl, 0.5% NP-40 supplemented with protease inhibitor cocktail). Bradford assay (BioRad) was performed to assay protein concentration and 20 µg of total protein was resolved on 4–12% Novex BOLT gradient gel, transferred to PVDF membrane and blotted with M73 hybridoma supernatant produced in-house to detect E1A or mouse pan-actin antibody (Fisher) as a loading control.

### RNA extraction

Total cellular RNA was extracted from infected cells 16 hours after infection using the TRIzol method as previously described [21]. Total RNA was then sent to Genome Québec for rRNA depletion, sequencing library preparation, and sequencing.

## RNA-Sequencing

RNA sequencing generated between 38 to 44 million paired reads per library using the Illumina Hiseq 2000/2500 sequencer. Libraries were prepared using the first strand TrueSeq mRNA protocol. Base calls were made using the Illumina CASAVA pipeline. Trimmomatic was used to remove Illumina sequencing adapters and to trim and clip low quality reads [22]. The reads were aligned to the *Homo sapiens* reference genome GRCh37 using the STAR aligner in a 2-pass mode [23]. Cufflinks were used to assemble aligned RNA-seq reads into transcripts and generate raw read counts [24]. Cufflinks and HTSeq were used to assemble aligned RNA-seq reads into transcripts to generate read counts [24–26]. Viral reads were aligned to a HAdV5 genome described by the Weitzman Lab available on GitHub (https://github.com/DepledgeLab/Ad5-annotation), which corrects and annotates the 2004 Genbank AC_000008.1 HAdV5 genome using nanopore long-read sequencing [27]. Adenovirus splicing variants were counted using the HTSeq 'nonunique fraction' setting due to overlap.

## Data analysis and visualization

Differential expression analysis (including normalization) was performed on raw reads using DESeq2 [28]. Genes with total counts less than 10 across all 10 samples were removed. The 'lfcshrinkage' function was used to reduce the variability of genes with low counts. Genes were designated as differentially expressed if they had a fold change of greater than |1.5| and an adjusted p value (BH) of less than 0.05. PCA charts were generated using pcaExplorer [29] with scaling via rlog transformation, heat plots with Pheatmap [30], and volcano plots with ggplot2 [31] and gene set enrichment analysis (GSEA) was performed with the clusterProfiler r packages [32]. Viral transcripts were graphed using GraphPad Prism (v10.4.1).

## Results

### E1A289R*dl*2–11/YC induces S-phase

We have previously observed that E1A is able to associate with the E2F/DP complex independently of its binding to Rb [2], this mutant (E1A289R*dl*2–11) was also able to drive S-phase in arrested primary lung fibroblasts similarly to E1A wild-type-expressing virus *dl*309 [2]. Our previous work also demonstrated that E1A289R is able to induce S-phase specific genes via direct recruitment to E2F4-occupied promoters, whereas E1A289R*dl*2–11 was unable to drive these genes. The low induction of S-phase specific genes by E1A289R*dl*2–11 but relatively efficient initiation of S-phase seems like a paradox; how does the virus expressing this E1A mutant drive S-phase? We wanted to initially determine whether removal of Rb binding had an effect on the ability of E1A289R*dl*2–11 to drive S-phase (Fig 1). Removal of Rb binding had no significant effect on the ability of E1A289R*dl*2–11/YC to drive S-phase, whereas it reduced the ability of E1A243R*dl*2–11/YC to induce S-phase significantly (Fig 1B). To better understand how E1A289R*dl*2–11/YC is able to induce S-phase, we undertook a transcriptomic analysis of how this virus reprograms the infected cell.

### Viruses expressing different E1A isoforms cluster together while Ad5.E1A289R*dl*2–11/YC is distinct in cellular transcriptome alteration

Initially, we verified that our mutant viruses express similar E1A protein levels when infected in arrested IMR-90 cells by western blot (Fig 2A). The levels of E1A proteins were broadly similar across the mutants investigated, with the expected banding pattern of E1A243R and E1A289R.

To investigate how E1A289R*dl*2–11/YC is able to re-program the infected cell and drive S-phase in arrested cells we performed RNA-seq on IMR-90 cells that were arrested for 72 hours by contact inhibition and then mock infected or infected with HAdV5 *dl*309 (wt E1A), *pm*975 (predominantly E1A 289R), *dl*520 (predominantly E1A 243R), or Ad5.E1A289R*dl*2–11/YC for 16 hours. Total RNA was then extracted from the cells using the Trizol method and was subjected

to sequencing using Illumina Hiseq 2000/2500 sequencer in duplicate. In total 23,289 genes were detected in at least one of the samples (Table 1). Differential expression analysis (DEA) was then performed using DESeq2. Out of those genes, we observed that 5955 were significantly altered (p-adjusted<0.05 and fold change of at least 1.5) in *dl*309 infected cells versus mock, 5297 in *pm*975 infected cells versus mock, 3039 in *dl*520 infected cells versus mock, and 6509 in Ad5. E1A289R*dl*2–11/YC-infected cells versus mock, with most of these genes upregulated (Table 1). Principal component analysis (PCA) showed that each duplicate clustered together, indicating that the biological replicates were similar (Fig 2B). Importantly, *dl*309, *pm*975 and *dl*520 samples all showed relative similarity to each other. Mock-infected cells were very distinct from other samples, as were cells infected with Ad5.E1A289R*dl*2–11/YC. A complete list of genes up and down regulated is available in the supplemental data.

A

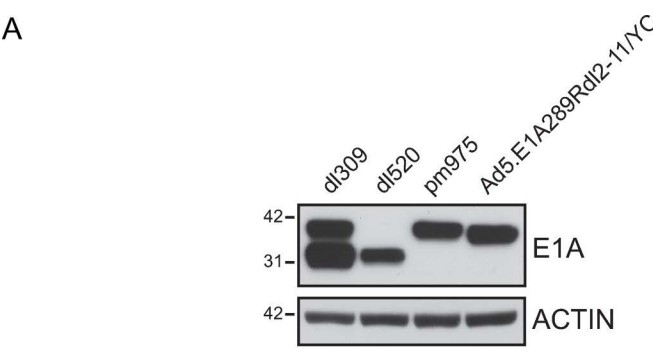

B

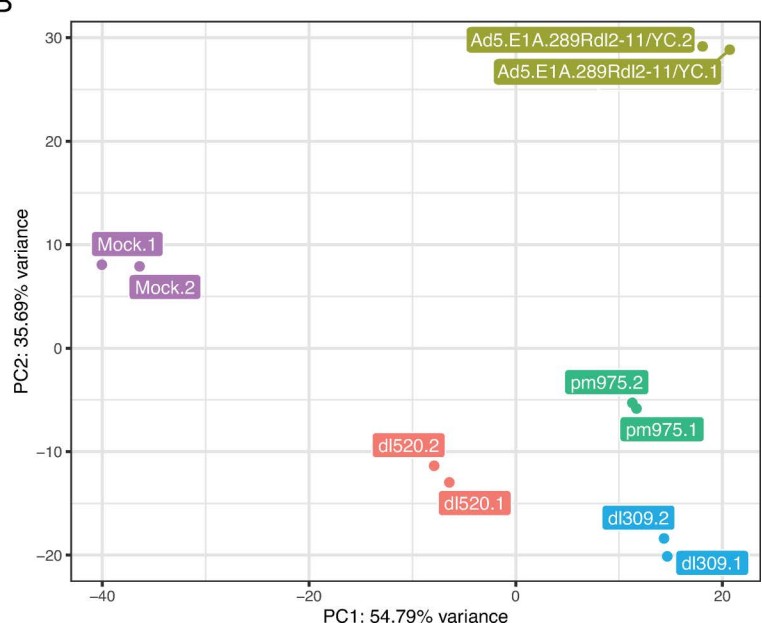

**Fig 2. E1A289R*dl*2-11/YC mutant is highly distinct from other groups.** A. Arrested IMR-90 cells were infected with the indicated HAdVs at an MOI of 10 for 16 hours after which the cells were harvested and 20 µg of total cell lysate was resolved on gradient 4-12% SDS-PAGE and blotted to PVDF. Lysates were probed with M73 anti-E1A antibody and actin as a loading control. Molecular weight markers, in kDa, are shown. B. RNA-Seq PCA. Principal Component Analysis plot generated using pcaExplorer with scaling via rlog transformation and displaying the first two principal coordinates. Individual replicates are colored according to sample (*dl*309, *pm*975, *dl*520, Ad5.E1A289R*dl*2-11/YC). Top 6385 most variable genes were used.

We also analyzed samples using references other than mock (Table 2). We observed that *pm*975 showed fewer cellular genes altered as compared to *dl*309 (362 genes) with most of those genes downregulated, while *dl*520 had significantly more altered genes (927 genes), with most of those also downregulated. Interestingly, Ad5.E1A289R*dl*2–11/YC had the most altered genes versus all other viruses, with comparison to *dl*309 showing 3429 altered genes, a 10-fold increase over *pm*975 in the same comparison. Together, these results indicate that despite significant changes to E1A binding partners, E1A289R*dl*2–11/YC is still able to reprogram the infected cell on a large scale. Supplemental Fig 1 shows Venn diagrams of overlapping genes for various comparisons.

### Ad5.E1A289R*dl*2–11/YC alters the cellular transcriptome in unique ways

To obtain a high-level picture of how the different viruses affect the infected cell we clustered the top 50 most variable genes across all samples on a heat map (Fig 3). We observed that the mock infected and the Ad5.E1A289R*dl*2–11/YC-infected samples were much more distinct from the remaining samples, which showed a relatively close clustering.

DEA by DESeq2 [28] was used for binary comparisons between the 5 samples (Fig 4). Looking at *dl*309 versus mock, many of the highly upregulated genes are involved in cell cycle regulation and stress response, with the most upregulated genes include *KISS1R, BTN1A1, NEXMIF, TP73*; while the most downregulated genes include *C10orf10, GALNT15, FMO2, and FAXDC2*. Most of the genes were upregulated in this sample. Similar results were observed in *pm*975 and *dl*520 infected samples as compared to mock, with many of the same genes highly up or down regulated as was observed in *dl*309-infected cells. Interestingly, several genes related to histone H1 were modestly downregulated in *dl*520 but not *dl*309 or *pm*975 infected cells. Furthermore, the number of differentially expressed genes in *dl*520 infected cells was somewhat lower compared to *dl*309 or *pm*975 infected cells. Lastly, comparison of Ad5.E1A289R*dl*2–11/YC to mock identified a number of significantly upregulated cytokine genes, including *CXCL5* and *CCL20*; stress response genes such as *HSPA1B, HSPA1A,* and *HSPA6*. Downregulated genes with this mutant included *FMO2, C10orf10, PPAP2B*.

**Table 1. RNA-Seq Samples. In total 23,289 genes were detected in at least one of the 5 samples. A total of 6385 genes had an adjusted pvalue (BH) of less than 0.05 and were up- or down-regulated as indicated in at least one of the 4 experimental samples using uninfected as a control.**

|  | Uninfected | *dl*309 | *pm*975 | *dl*520 | Ad5.E1A289R*dl*2–11/YC |
|---|---|---|---|---|---|
| *Base Counts* | 22081 | 22267 | 21736 | 21914 | 21753 |
| *Fold Change > 2* |  | 3059 | 2453 | 1118 | 3419 |
| *Upregulated* |  | 2072 | 1620 | 846 | 1893 |
| *Downregulated* |  | 987 | 833 | 272 | 1526 |
| *Fold Change > 1.5* |  | 5955 | 5297 | 3039 | 6509 |
| *Upregulated* |  | 3553 | 3135 | 1931 | 3545 |
| *Downregulated* |  | 2402 | 2162 | 1162 | 2964 |

**Table 2. RNA-Seq Sample Comparisons (adjusted pvalue < 0.05).**

|  | *pm*975/ *dl*309 | *dl*520/*dl*309 | *pm*975/*dl*520 | Ad5.E1A289R*dl*2–11/ YC/*dl*309 | Ad5.E1A289R*dl*2–11/ YC/*pm*975 | Ad5.E1A289R*dl*2–11/ YC/*dl*520 |
|---|---|---|---|---|---|---|
| *Fold Change > 2* | 81 | 249 | 282 | 1669 | 822 | 1735 |
| *Upregulated* | 11 | 8 | 233 | 679 | 373 | 869 |
| *Downregulated* | 70 | 241 | 49 | 990 | 449 | 866 |
| *Fold Change > 1.5* | 362 | 927 | 973 | 3429 | 1988 | 3890 |
| *Upregulated* | 105 | 134 | 647 | 1449 | 906 | 1856 |
| *Downregulated* | 257 | 793 | 326 | 1980 | 1082 | 2034 |

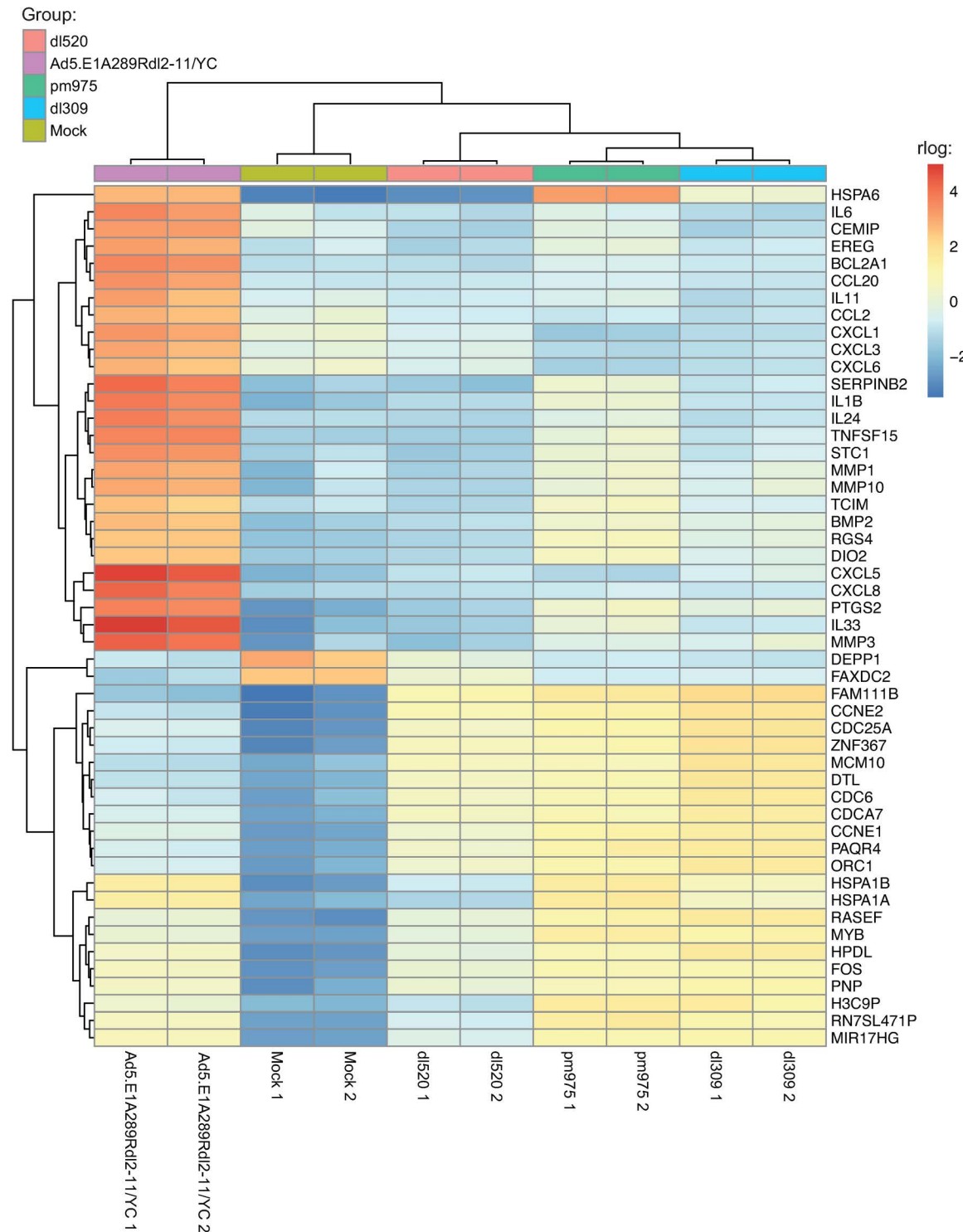

**Fig 3. Heat shock proteins and cytokines are among the most variable transcripts across samples.** RNA-Seq Heat Map of the 50 most variable genes among all 5 samples. Heat map generated using pheatmap with hierarchical clustering (samples and transcripts) after using DESeq2 rlog transformation for scaling. Boxes in red are upregulated, blue are downregulated. Dendrograms indicate similarity, i.e., less branches between two samples (or genes) means higher similarity.

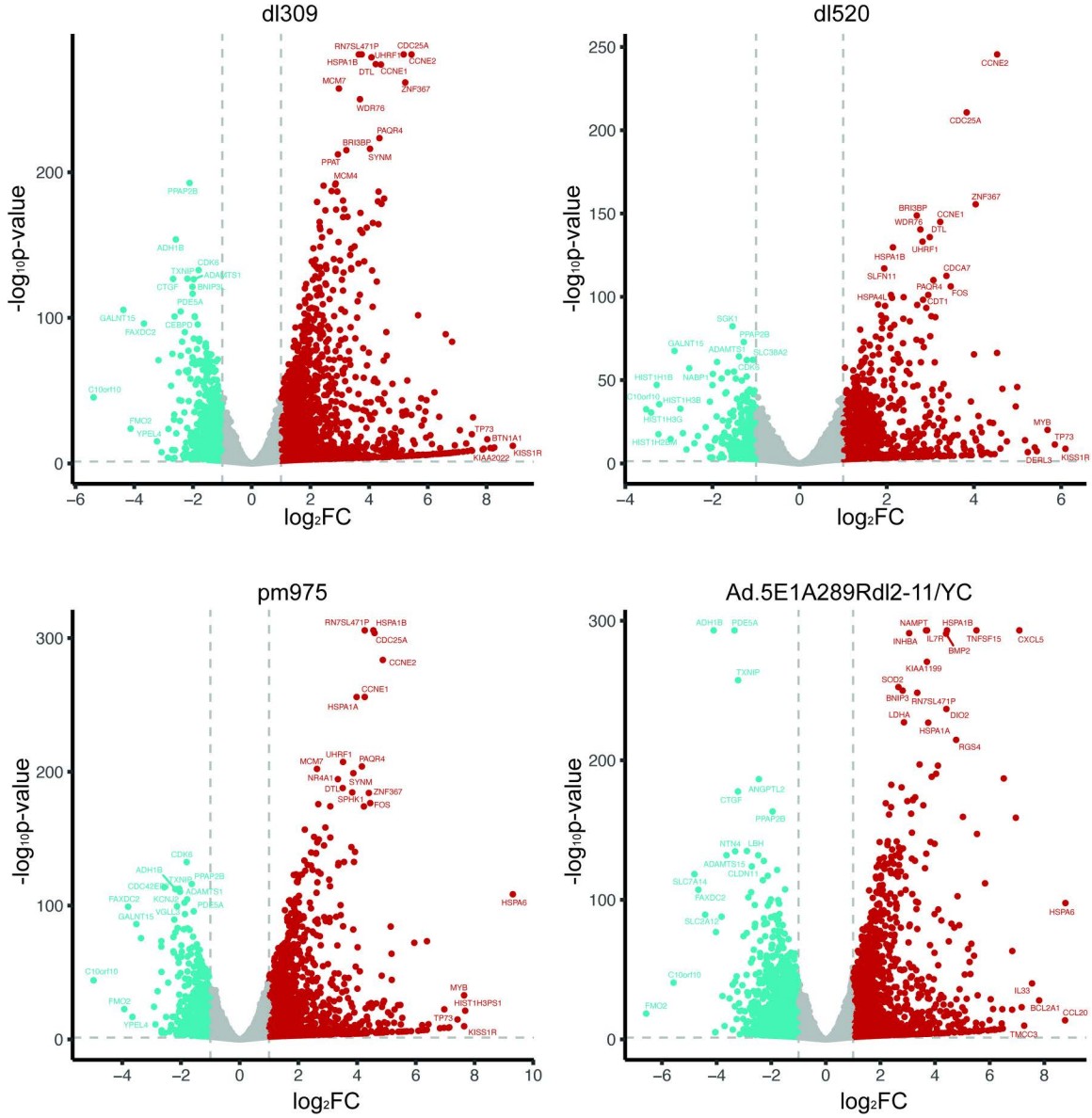

**Fig 4. Volcano Plots showing differential expression of transcripts in *dl*309, *pm*975, *dl*520, and Ad5. E1A289R*dl*2-11/YC-infected cells compared to mock-infected cells.** DESeq2 was used for differential expression analysis including p-values and fold changes. Red indicates upregulated (FC > 1.5, adjusted pvalue < 0.05), blue indicates downregulated (FC < -1.5, pvalue < 0.05). Volcano plot generated using ggplot2.

We also wanted to understand how the mutants compared to *dl*309-infected samples in their alteration of the cellular transcriptome (Fig 5). Comparison of *pm*975 to *dl*309 revealed comparatively few genes that were significantly changed, with the overall level of alteration in expression being relatively low. Notable upregulated genes include *C8orf48, NEB, KLH3, C8orf4* and *IL1B*. In terms of downregulated genes, notable genes include *RAD54L, KIF2C, E2F2*, and *BLM*. Comparing *dl*520 to *dl*309 revealed that very few genes were upregulated, with those that were showing a very modest level such as *FMO2, C10orf10, TXNIP,* and *GALNT15*. There were many more genes that were downregulated and notable ones include *HSPA6, RN7SL832P, NKX2–4,* and *RGS16*. Comparing Ad5.E1A289R*dl*2–11/YC to *dl*309 revealed

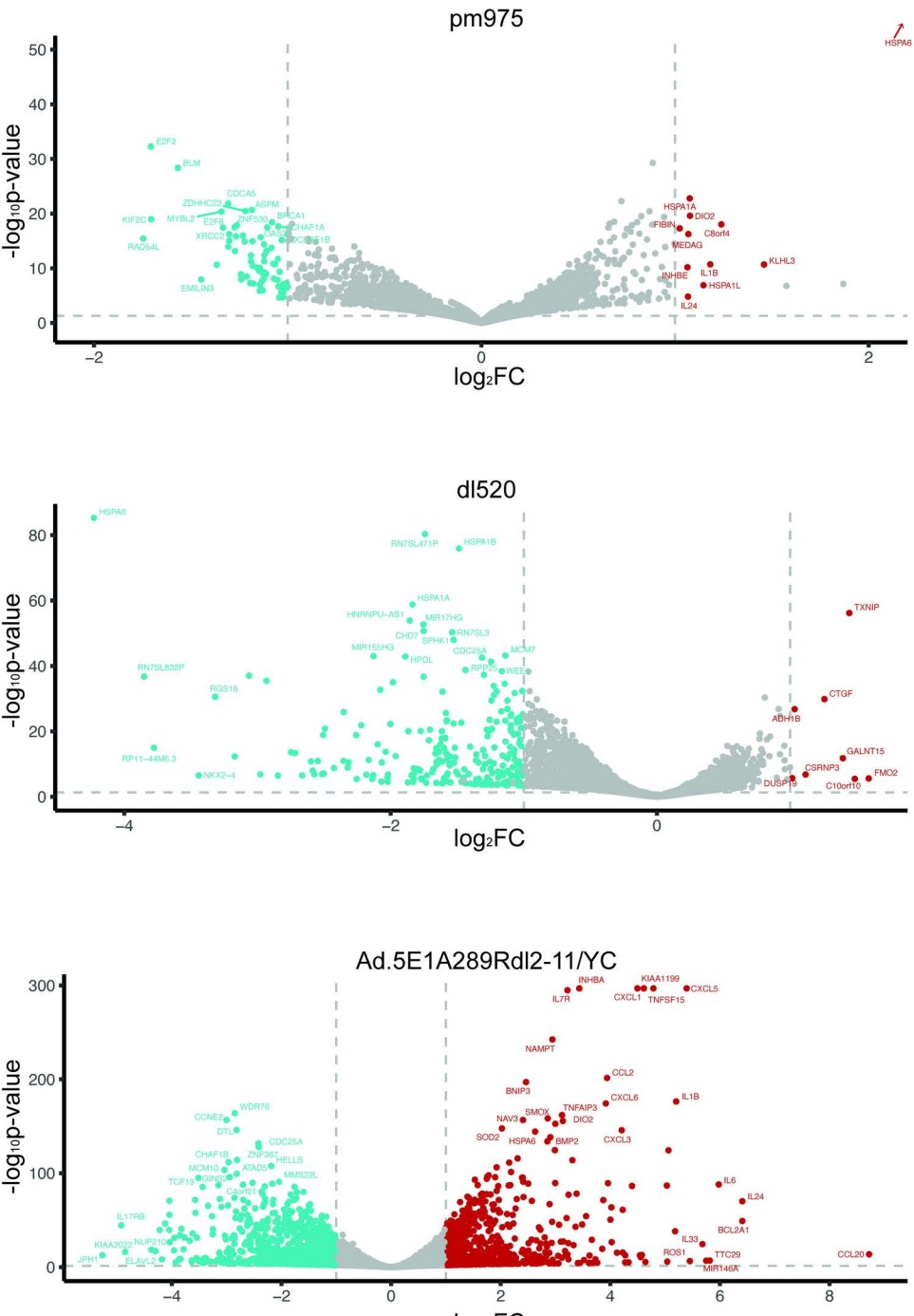

**Fig 5. Volcano Plots showing differential expression of transcripts in *pm*975, *dl*520, and Ad.5E1A289R*dl*2-11/YC compared to *dl*309.**
DESeq2 was used for differential expression analysis including p-values and fold changes. HSPA6 (-log$_{10}$(padj)=200, log$_2$FC=3.11) was removed from *pm*975/*dl*309 for visualization purposes. Red indicates upregulated (FC>1.5, adjusted pvalue<0.05), blue indicates downregulated (FC<-1.5, pvalue<0.05). Volcano plot generated using ggplot2.

a relatively large number of genes that were up or down regulated, some showing relatively high levels of change in expression not observed with other mutants. Notable genes upregulated were *CCL20, IL24, IL6, BCL2A1, TTC29*. Genes downregulated include *JPH1, KIAA2022, IL17RB, NUP210*, and *ELAVL2*.

Lastly, we compared Ad5.E1A289R*dl*2–11/YC to *pm*975 and *dl*520. E1A289R*dl*2–11/YC over E1A289R showed many genes up and down regulated (Supplemental Fig 2). Upregulated genes include *CA9, CCL20, mIR146A, CXCL5,* and *BCL2A1*. Downregulated genes include *KIAA2022, IL17RB, JPH1, FAM11B,* and *GNAZ*. Comparing the mutant to *dl*520, we observed a large number of up and down regulated genes, many of the same genes we observed in other comparisons. Upregulated genes include *BCL21A, CCL20, IL24, IL3, MMP12, HSPAA6*; while downregulated genes include *FMO2, FAM211A, JHP,* and *KHK*. Overall, Ad5.E1A289R*dl*2–11/YC had the largest number of unique genes compared to other samples.

Expression of viral genes was also analyzed with most viral genes expressed at a similar level (Supplemental Fig 3), with an overall trend of reduced viral early gene expression observed in *dl*520 and Ad5.E1A289R*dl*2–11/YC. Expression of most late genes was predominantly reduced in *dl*520, with Ad5.E1A289E*dl*2–11/YC showing expression comparable to *dl*309 (Supplemental Fig 3).

Lastly, we include alternate volcano plots of Figs 4 and 5 with the Y-axis maintained across all plots to allow for direct comparison. This is presented as Supplemental Figs 4 and 5.

## Functional analysis of altered pathways

To gain a better grasp on the functional pathways within the infected cell altered by the infection of different viruses we performed GSEA of GO categories (Fig 6). Infection of cells with *dl*309 resulted in significant enrichment of many pathways in the cell as compared to uninfected cells, with the most enriched terms being DNA replication, nuclear chromosome, cell cycle DNA replication, ribosome biogenesis, a variety of nucleic acid metabolic pathways, and stress response. Downregulated terms included a variety of cellular differentiation pathways, immune responses, cell adhesion, and cell polarity. Comparing *pm*975 to uninfected cells resulted in a similar distribution of GO terms as was observed for *dl*309 for upregulated pathways, with stress response terms being more prominent. Downregulated pathways were somewhat different and included small GTPase mediated signal transduction, regulation of the Wnt signalling pathway, autophagosome, Rho protein signal transduction, along with cellular differentiation and immune response terms. Comparisons of *dl*520 to uninfected cells identified many downregulated pathways as compared to *dl*309 and *pm*975, with fewer pathways upregulated. The upregulated pathways were similar to those observed for *dl*309 and *pm*975, and included DNA replication, cell cycle, and DNA repair. While downregulated pathways included cell polarity, cell adhesion, differentiation pathways, and immune response pathways. Lastly, comparing Ad5.E1A289R*dl*2–11/YC to mock-infected cells identified significant number of upregulated pathways, including stress response, ribosome biogenesis, immune response, GPCR binding, and cell cycle. Downregulated pathways included cell adhesion, cytoskeleton, and extracellular matrix.

To obtain a more unique comparison of how Ad5.E1A289R*dl*2–11/YC affects the infected cells, we performed GSEA of this mutant compared to the remaining three mutant viruses with *dl*309 being considered phenotypically wild-type (Supplemental Fig 6). Comparing Ad5.E1A289R*dl*2–11/YC to *dl*309 identified a number of upregulated pathways, including immune pathways, receptor signalling pathways, and apoptotic pathways. Downregulated pathways included DNA replication, DNA repair, chromosome organization, mitotic cell cycle, and ubiquitin ligase complex. Comparing this mutant to *pm*975 identified mostly upregulated pathways, including immune pathways, stress response pathways, growth factor activity, and locomotion. Downregulated pathways consisted of DNA replication and cell cycle. Finally, comparing this mutant to *dl*520 identified mostly upregulated pathways, including immune pathways, stress response, and cell growth. Downregulated pathways include cell cycle and DNA replication. A complete list of enriched GO terms for each comparison is included as supplemental data.

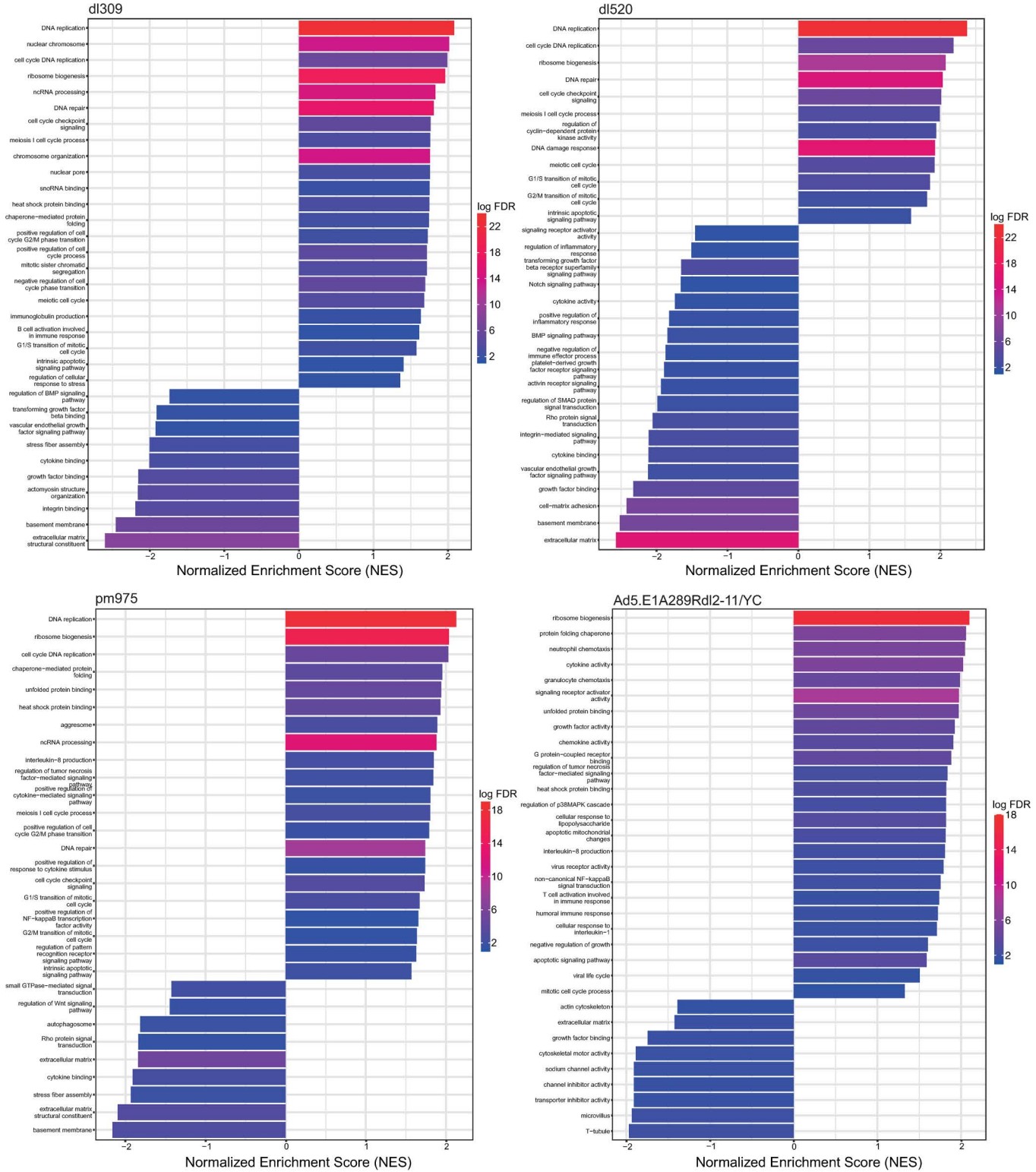

**Fig 6. GSEA GO Charts of *dl*309/mock, *pm*975/mock, *dl*520/mock, and Ad5. E1A289R*dl*2-11/YC/mock.** GSEA of differential expression comparisons. The x axis is the normalized expression score (NES), indicating whether GO terms (y axis) are upregulated (NES > 0) or downregulated (NES < 0). Bars are colored based on logFDR (false discovery rate) (scale bar on side). GSEA and bar charts generated using clusterProfiler R package.

## Discussion

In the present study we investigated the impact of different HAdV mutants on the cellular transcriptome with the goal of understanding how these viruses differentially reprogram the infected cell and how they drive these cells into the S-phase. Arrested, via a 72 hour contact inhibition, primary lung fibroblasts IMR-90 were infected with each virus mutant at an MOI of 10 for 16 hours to capture early cellular reprogramming. The reasons for this study stemmed from our observation that an E1A mutant unable to bind to either the E2F/DP complexes via the N-terminus residues 2–11 or to Rb-family of proteins was still able to efficiently drive S-phase (Figs 1 and 2). We have identified numerous differences between the viral mutants that shed light on the mechanisms of cell cycle deregulation, immune response, and transcriptomic reprogramming by each E1A isoform. Interestingly, viruses expressing the different, unmutated, E1A isoforms were relatively similar, while Ad5.E1A289R*dl*2–11/YC was dramatically different, and its reprogramming was therefore rather unique.

Most prior transcriptomic studies of adenovirus infected cells have focused on the wild-type virus, rather than mutants expressing mutated or specific E1A isoforms [33–38]. Our study is unique in that it investigated the transcriptional changes induced by each early E1A isoform individually and compared it to a mutant that would be expected to be largely deficient for S-phase induction (E1A289R*dl*2–11/YC).

Induction of the cell cycle is a necessity for the virus to replicate and hence HAdVs have evolved sophisticated mechanisms of ensuring that this occurs, previously reviewed by [39–41]. Our previous work identified that E1A can directly bind to the E2F/DP complex to drive E2F-regulated gene expression directly, rather than relying exclusively on disruption of Rb (and related) protein function through CR1 and CR2-mediated interactions [2]. Infection of the arrested cell by *dl*309, which is considered phenotypically wild-type in cell culture, mimicked transcriptional changes previously reported in the same IMR-90 cells infected with HAdV-C2 [33] as we have used here. Viruses expressing individual E1A isoforms behaved distinctly, albeit within the constraints of what was observed for *dl*309. The mutant *pm*975, expressing E1A289R and E1A217R, although mainly E1A289R would be expressed this early in infection (Fig 2A) [42], showed relatively few genes differentially regulated compared to *dl*309 and over mock-infected cells. This is consistent with our prior observations that *pm*975 is phenotypically very similar to *dl*309 in arrested IMR-90 cells [42]. Indeed, our previous work [42] showed that this mutant grows to slightly higher titers than *dl*309, and consistent with this, it expressed some viral genes to a slightly higher level (Supplemental Fig 3) resulting in higher overall genome replication as compared to *dl*309 [42]. The mutant *dl*520, expressing predominantly E1A243R and E1A171R, with mainly E1A243R expressed at 16 hours after infection (Fig 2A) [42], was previously shown to be deficient in growth, gene expression, and viral gene activation as compared to *dl*309 [42] and we observed that viral gene expression was also reduced in the current study for many viral genes (Supplemental Fig 3). However, early in infection the deregulation of the cellular transcriptome by HAdV expressing only these isoforms was not dissimilar to what we have observed with *dl*309 and *pm*975. Yet, when compared directly to *dl*309, *dl*520 showed relatively few genes expressed at higher levels than *dl*309 or *pm*975, with most genes expressed at lower levels. This is consistent with our previous observations, where E1A243R is considered a transcriptional repressor and functions together with E1A289R to optimally reprogram the infected cell for maximal viral replication [2,10]. The effects of the other E1A isoforms (211R, 171R, and 55R) are unlikely to be significant at this early stage of infection as they are not expressed at any detectable levels this early in IMR-90 cells (Fig 2A) [42].

The question of how E1A289R*dl*2–11/YC is able to drive S-phase in the absence of direct binding to the E2F/DP complex and Rb proteins is further illuminated by our results. Compared to *pm*975, Ad5.E1A289R*dl*2–11/YC does not drive expression of cell cycle genes to the same level, and many are expressed at levels low enough that they are not statistically enriched/overexpressed in GSEA. Still, several cell cycle genes were significantly upregulated as compared to mock-infected cells by E1A289R*dl*2–11/YC-expressing mutant, including *CCNE1, CCNE2, CDCA7, ORC1, MCM7, MCM10,* and *MCM3*; while others were modestly upregulated such as *E2F*s. In the absence of known S-phase pathways that E1A uses to drive S-phase, it is unclear how these genes get activated. One possibility could be that MYC is driving S-phase. MYC is known to be mitogenic and has previously been shown to drive S-phase and activate E2F-regulated genes [43,44]. Levels of *MYC* transcripts were the highest in E1A289R*dl*2–11/YC expressing cells at approximately

5.3-fold higher than in mock-infected cells, this is significantly higher than what was observed in *dl*309-infected cells where the upregulation was only 1.6-fold, or *pm*975-infected cells where this was 3-fold. In *dl*520, *MYC* transcripts were downregulated to 0.73-fold of mock-infected cells. MYC-mediated S-phase activation would normally lead to apoptosis through either activation of cytochrome C release [45] or stimulation of the ARF/p53 pathway [46]. Although our experiment was only performed for 16 hours, making it impossible to assess whether apoptosis would occur through cytochrome C release, we did not detect significant upregulation of either *TP53* or *CDKN2A* in Ad5.E1A289R*dl*2–11/YC-infected cells. Nevertheless, further studies could shed light on whether this occurs. Another possibility involves the viral E4 transcriptional unit. Previous studies have shown that E4orf6/7 is able to drive E2F4 into the nucleus and activate E2F-regulated promoters as well as induce the expression of E2F1 [13,15,47]. Our results show that the levels of viral E4 transcripts are relatively similar across all viruses (Supplemental Fig 3), suggesting that their expression is not impaired and in the absence of canonical S-phase induction pathways via E1A, E4 proteins, particularly E4orf6/7, may compensate to provide an alternate S-phase induction mechanism. Importantly, E4 transcription is initiated at nearly the same time as E1A in normal cells [48], suggesting that it may initially be activated independently of E1A and raising a warning flag for vectors that are E1-deleted in hopes of preventing E4 activation. Ultimately, it is very likely that both MYC and E4orf6/7 contribute to S-phase induction and future studies should be able to answer this question more thoroughly.

Lastly, the strong induction of interferon stimulated genes (ISGs) and various cytokines hints at possible mechanisms by which E1A is able to suppress the expression of these molecules during infection. Apart from Ad5.E1A289R*dl*2–11/YC, the viruses did not induce a strong innate immune response in the infected cells based on our transcriptomic results. These observations suggest that either the N-terminal residues 2–11 of E1A or Rb binding is important for suppression of innate immune pathways. Deletion of residues 2–11 will impair binding to p300/CBP [10], which associates with IRF3 to drive ISG activation [49]. This region is a known transcriptional repression domain of E1A [11], and E1A has previously been shown to suppress ISGs [50] via a p300/CBP-dependent mechanism [51]. Therefore, it is likely that the elevated expression of innate immune genes by E1A289R*dl*2–11/YC is caused by the inability of this mutant to interact with p300/CBP and possibly other mechanisms. Indeed, p300/CBP is rather limited in cells [10] and inability to further sequester it could drive ISG activation.

The present study investigated the transcriptomic differences between cellular genes affected by HAdV5 mutants expressing different E1A isoforms. We demonstrate that although the viruses affect the cellular transcriptome in similar ways, there are many significant differences. Significantly, we show that ablation of the normal regions of E1A thought essential for S-phase induction is not necessary to drive infected cells into the S-phase. Similarly, we show that deletion of Rb and E2F/DP binding regions (along with p300/CBP binding regions that overlaps E2F/DP binding region in the N-terminus of E1A) makes the virus unable to efficiently suppress innate immune pathways. These findings provide further insight into molecular reprogramming of the infected cell and offer important caveats for those using adenovirus for oncolytic virotherapy or other therapies. In particular, the mechanisms that the viruses use to drive S-phase even in the absence of Rb-family protein binding domains, making such oncolytic viruses less selective than otherwise would be the case.

## Supporting information

**Supplemental Fig 1. RNA-Seq Venn Diagrams (A) compared to *dl*309 (B) compared to mock. Venn diagrams generated using ggVennDiagram, displaying the amount of differentially expressed transcripts (FC > 2, adjusted pvalue < 0.05).**
(PDF)

**Supplemental Fig 2. Volcano Plots showing differential expression of transcripts in Ad5.E1A289R*dl*2-11/YC compared to *pm*975 and *dl*520. DESeq2 was used for differential expression analysis including p-values and fold changes. Red indicates upregulated (FC > 1, pvalue < 0.05), blue indicates downregulated (FC < -1, pvalue < 0.05). Volcano plot generated using ggplot2.**
(PDF)

**Supplemental Fig 3. Viral Gene Expression.** Bar charts showing normalized expression levels of viral transcripts for mock (light blue), *dl*309 (purple), *pm*975 (magenta), *dl*520 (green), and Ad5.E1A289R*dl*2-11/YC (dark blue). Counts were normalized using DESeq2 and scaled using a $\text{Log}_2$ transformation.
(PDF)

**Supplemental Fig 4. Volcano Plots** showing differential expression of transcripts in *dl*309, *pm*975, *dl*520, and Ad5. E1A289R*dl*2-11/YC-infected cells compared to mock-infected cells. DESeq2 was used for differential expression analysis including p-values and fold changes. Red indicates upregulated (FC > 1.5, adjusted pvalue < 0.05), blue indicates downregulated (FC < -1.5, pvalue < 0.05). Volcano plot generated using ggplot2.
(PDF)

**Supplemental Fig 5. Volcano Plots** showing differential expression of transcripts in *pm*975, *dl*520, and Ad.5E1A289R*dl*2-11/YC compared to *dl*309. DESeq2 was used for differential expression analysis including p-values and fold changes. HSPA6 ($-\log_{10}$(padj)=200, $\log_2$FC=3.11) was removed from *pm*975/*dl*309 for visualization purposes. Red indicates upregulated (FC > 1.5, adjusted pvalue < 0.05), blue indicates downregulated (FC < -1.5, pvalue < 0.05). Volcano plot generated using ggplot2.
(PDF)

**Supplemental Fig 6. GSEA Bar Charts** of Ad.5E1A289R*dl*2-11/YC/309, Ad5.E1A289R*dl*2-11/YC/*pm*975, and Ad5.E1A289R*dl*2-11/YC/*dl*520. GSEA of differential expression comparisons. The x axis is the NES, indicating whether GO terms (y axis) are upregulated (NES > 0) or downregulated (NES < 0). Bars are colored based on logFDR (false discovery rate) (scale bar on side). GSEA and bar charts generated using clusterProfiler R package.
(PDF)

**Supplemental Excel Spreadsheets.** Excel spreadsheets with full differential expression tables and GSEA tables.
(ZIP)

## Acknowledgments

We thank Dr. Marike Palmer for technical assistance. PP thanks Stanisława Pełka for invaluable support and assistance and Ryszard Pełka for inspiration and curiosity, and for their critical feedback on the manuscript.

## Author contributions

**Conceptualization:** Rafe Helwer, Peter Pelka.

**Data curation:** Peter Pelka.

**Formal analysis:** Rafe Helwer, Peter Pelka.

**Funding acquisition:** Peter Pelka.

**Methodology:** Rafe Helwer.

**Project administration:** Peter Pelka.

**Supervision:** Peter Pelka.

**Visualization:** Rafe Helwer, Peter Pelka.

**Writing – original draft:** Peter Pelka.

**Writing – review & editing:** Rafe Helwer, Peter Pelka.

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
