## [Decision Letter · Decision Letter 0]

27 Mar 2025

PONE-D-25-09538Cellular transcriptomics of arrested normal lung fibroblasts IMR-90 infected with human adenovirus 5 E1A mutantsPLOS ONE

Dear Dr. Pelka,

Thank you for submitting your manuscript to PLOS ONE. After careful consideration, we feel that it has merit but does not fully meet PLOS ONE’s publication criteria as it currently stands. Therefore, we invite you to submit a revised version of the manuscript that addresses the points raised during the review process.

<!--StartFragmentBoth reviewers agreed that your manuscript provides valuable and new descriptive information on the changes provoked by adenovirus infection. however, there are a few issues raised by the reviewers that need to be addressed carefully. Please see reviewers' insightful comments below.

We look forward to receiving your revised manuscript.

Kind regards,

Baochuan Lin, Ph.D.

Academic Editor

PLOS ONE

Journal Requirements:

2**. ** Thank you for stating the following financial disclosure:

“This work was supported by grants from the Natural Sciences and Engineering Research Council of Canada to PP (Grant number: RGPIN/05366-19) and the Canadian Institutes of Health Research to PP (Grant number: PJT-173376). Funding for open access charge: CIHR PJT-173376. RH was supported with a studentship from Research Manitoba.”

Reviewers' comments:

Reviewer's Responses to Questions

**Comments to the Author**

1. Is the manuscript technically sound, and do the data support the conclusions?

Reviewer #1: Partly

Reviewer #2: Yes

2. Has the statistical analysis been performed appropriately and rigorously? 

Reviewer #1: Yes

Reviewer #2: Yes

3. Have the authors made all data underlying the findings in their manuscript fully available?

Reviewer #1: Yes

Reviewer #2: Yes

4. Is the manuscript presented in an intelligible fashion and written in standard English?

Reviewer #1: Yes

Reviewer #2: Yes

5. Review Comments to the Author

Reviewer #1: HAdVC-5 E1A protein has been extensively studied as it is the first protein expressed following infection and has many functions to result in efficient virus life cycle. Many protein binding domains have been identified, and probably more will be in the future. Major E1A interacting proteins are pRB (through the CR1 and CR2 motifs) and E2F (through aa 2-11). Theses interactions are thought to be critical for induction of S phase and viral DNA replication. In this manuscript, the authors have found that a virus with an E1A mutated in both these domains was still able to induce cell cycle in infected arrested cells. In an attempt to understand how the mutant virus accomplishes this, a transcriptomic analysis was performed comparing RNA expression of different mutant viruses. The results obtained showed a very surprising effect when both sites were mutated, with more changes even that the WT viruses. Follow up analysis of this data could provide interesting insights into more roles of E1A during infection.

The data seems to be properly analysed and is well presented. The manuscript is well written and easy to follow. The conclusions made from the results are mostly well supported by the data. However, the supplementary Fig. 4 was not provided. From the description in the text, this figure would be important, and even probably should be part of the main figures. It is thus unclear which western blots were done for protein expression. Of particular necessity is the expression levels of the different E1A proteins to ascertain that the different effect seen on transcription are not simply due to big difference in expression levels. It would also be interesting to have expression levels of additional viral proteins, especially the early proteins E4orf6 and E1B55K, as described below in the first point of the minor issues

Minor issues

- In the introduction (line 51), the authors mention other viral protein can contribute to the dysregulation of the cell cycle in infected cells. They have however forgotten to add the ligase complex formed by the E4orf6 protein with the E1A55K as shown by Dallaire et al. 2015 mSphere 1 (1).

- Line 73 That sentence is confusing, should be reworded.

- Fig. 2 (line 143) indicates that significant expression alteration is a change of more than 2 whereas the Materials and Methods section indicates a change of more than1.5. It should be consistent.

- Fig. 4 The fourth panel is not described in the text.

- The Material and methods section indicates that viral transcripts were also analysed. It would be interesting to also show and discuss these.

Reviewer #2: The report of Helwer and Pelka (PONE-D-25-09538) is an informative and valuable contribution to our understanding of how adenovirus reprograms the cell to favor viral DNA replication. This analysis of adenovirus mutants that differ in the expressed form of the major immediate early gene products shows new insight into mechanisms engaged by the virus. An even more significant contribution is that that the canonical pathway of inactivating the tumor suppressor pRb in order promote the transcription of S-phase associated genes is not the only means of provoking this change in the resting cell. Moreover, the analysis of differentially expressed genes provided by this study suggest the “alternative” pathway may involve c-Myc, with ancillary changes in the expression of genes associated with the innate immune response and cell-cell interactions.

Notable strengths of this report include the clear and informative review of pertinent literature, including an underappreciated recognition that other products expressed by adenovirus such as E4orf6/7 can promote entry into an S-phase-like state. The comparisons were made among cells infected with viruses expressing either the wild-type, large (13S or 289R), small (12S or 243R) or mutant large E1A product with a deletion and the inability to bind pRb. The diploid fibroblasts chosen for these study provide the benefit of being able to be growth-arrested even though they are not a significant target for adenovirus infection in the human. The methods are clearly described and presented. With one exception, noted below, the results are clear and unambiguous and support the thoughtful discussion.

In summary this report provides valuable and new descriptive information on the changes provoked by adenovirus infection. The use of critical viral mutants provides new insight into the potentially complex pathways by which the virus interacts with the host. Even though the primary fibroblast may not be the most “native” target for adenovirus, the results provided here can readily be applied or extended in other studies examining transformation or adenovirus disease.

I raise two issues of cosmetic concern regarding the volcano plots in Figs. 4 and 5. First, the font size used to label significant genes is far too small to be of any use in a printed copy of the report. Perhaps fewer genes could be labeled, or the reader can simply refer to the sorted data. Second, in my opinion, the limits of the x axis in Fig. 5 should be made identical. Since I recommend eliminating the labeled genes, by harmonizing the scale between the three panels, the striking absence of an effect for the 12S- and 13S-expressing viruses compared to the 13S-mutant becomes much more obvious. That seems to be more important message to convey than the identification of genes with marginal significance.

6. PLOS authors have the option to publish the peer review history of their article (what does this mean? ). If published, this will include your full peer review and any attached files.

**Do you want your identity to be public for this peer review?** For information about this choice, including consent withdrawal, please see our Privacy Policy .

Reviewer #1: No

Reviewer #2: **Yes: ** David A. Ornelles

---

## [Author Response · Author response to Decision Letter 1]

1 Apr 2025

We wanted to thank the reviewers for their hard work and insightful comments. We have addressed all of the reviewers comments and our point-by-point response is provided below. Original reviewer comments are in quotations.

Reviewer #1.

“Of particular necessity is the expression levels of the different E1A proteins to ascertain that the different effect seen on transcription are not simply due to big difference in expression levels. It would also be interesting to have expression levels of additional viral proteins, especially the early proteins E4orf6 and E1B55K, as described below in the first point of the minor issues”

Response: We have now included the requested E1A blot as Figure 2A showing similar levels of E1A protein across all viruses. We have attempted to blot for E4orf6 and E1B-55k but were unable to obtain satisfactory results as these antibodies are not as robust as the E1A antibody and expression is expected to be low at 16 hours after infection. Nevertheless, viral gene expression is shown in Supplemental Figure 3 showing comparable levels of these genes mRNAs in the viruses examined.

“In the introduction (line 51), the authors mention other viral protein can contribute to the dysregulation of the cell cycle in infected cells. They have however forgotten to add the ligase complex formed by the E4orf6 protein with the E1A55K as shown by Dallaire et al. 2015 mSphere 1 (1).”

Response: We apologize for the omission, and this reference has now been incorporated into the introduction.

“Line 73 That sentence is confusing, should be reworded.”

Response: This has been reworded.

“Fig. 2 (line 143) indicates that significant expression alteration is a change of more than 2 whereas the Materials and Methods section indicates a change of more than 1.5. It should be consistent.”

Response: This was changed to be consistent.

“Fig. 4 The fourth panel is not described in the text.”

Response: The description has not been added to the results.

“The Material and methods section indicates that viral transcripts were also analysed. It would be interesting to also show and discuss these.”

Response: This is now included as Supplemental Figure 3.

Reviewer #2:

“I raise two issues of cosmetic concern regarding the volcano plots in Figs. 4 and 5. First, the font size used to label significant genes is far too small to be of any use in a printed copy of the report. Perhaps fewer genes could be labeled, or the reader can simply refer to the sorted data. Second, in my opinion, the limits of the x axis in Fig. 5 should be made identical. Since I recommend eliminating the labeled genes, by harmonizing the scale between the three panels, the striking absence of an effect for the 12S- and 13S-expressing viruses compared to the 13S-mutant becomes much more obvious.”

Response: We have now increased the size of the text labels on the volcano plots (Figures 4 and 5) and have included versions of volcano plots with harmonized Y-axes as suggested by the reviewer as Supplemental Figures 4 and 5.

We hope that the changes we have made will make our manuscript suitable for publication in PLOS One.

Thank you,

The authors.

---

## [Decision Letter · Decision Letter 1]

9 Apr 2025

Cellular transcriptomics of arrested normal lung fibroblasts IMR-90 infected with human adenovirus 5 E1A mutants

PONE-D-25-09538R1

Dear Dr. Pelka,

We’re pleased to inform you that your manuscript has been judged scientifically suitable for publication, Please correct one typo on line 56, change E3orf6 to E4orf6. The manuscript will be formally accepted for publication once it meets all outstanding technical requirements.

Kind regards,

Baochuan Lin, Ph.D.

Academic Editor

PLOS ONE

Additional Editor Comments (optional):

Reviewers' comments:

Reviewer's Responses to Questions

**Comments to the Author**

1. If the authors have adequately addressed your comments raised in a previous round of review and you feel that this manuscript is now acceptable for publication, you may indicate that here to bypass the “Comments to the Author” section, enter your conflict of interest statement in the “Confidential to Editor” section, and submit your "Accept" recommendation.

Reviewer #1: All comments have been addressed

Reviewer #2: All comments have been addressed

2. Is the manuscript technically sound, and do the data support the conclusions?

Reviewer #1: Yes

Reviewer #2: (No Response)

3. Has the statistical analysis been performed appropriately and rigorously? 

Reviewer #1: Yes

Reviewer #2: (No Response)

4. Have the authors made all data underlying the findings in their manuscript fully available?

Reviewer #1: Yes

Reviewer #2: (No Response)

5. Is the manuscript presented in an intelligible fashion and written in standard English?

Reviewer #1: Yes

Reviewer #2: (No Response)

6. Review Comments to the Author

Reviewer #1: There is one additional typo in the added text to the revised manuscript.

In line 56 it is written E3orf6 where it should be E4orf6

Reviewer #2: (No Response)

7. PLOS authors have the option to publish the peer review history of their article (what does this mean? ). If published, this will include your full peer review and any attached files.

**Do you want your identity to be public for this peer review?** For information about this choice, including consent withdrawal, please see our Privacy Policy .

Reviewer #1: No

Reviewer #2: **Yes: ** David Ornelles

---

## [Editor Report · Acceptance letter]

PONE-D-25-09538R1

PLOS ONE

Dear Dr. Pelka,

I'm pleased to inform you that your manuscript has been deemed suitable for publication in PLOS ONE. Congratulations! Your manuscript is now being handed over to our production team.

Kind regards,

on behalf of

Dr. Baochuan Lin

Academic Editor

PLOS ONE